# Limited impact of hydrogen co-firing on prolonging fossil-based power generation under low emissions scenarios

**Ken Oshiro** [1] ✉ **& Shinichiro Fujimori** [1,2,3]

Climate change mitigation generally require rapid decarbonization in the power sector, including phase-out of fossil fuel-fired generators. Given recent technological developments, co-firing of hydrogen or ammonia, could help decarbonize fossil-based generators, but little is known about how its effects would play out globally. Here, we explore this topic using an energy system model. The results indicate that hydrogen co-firing occurs solely in stringent mitigation like 1.5 °C scenarios, where around half of existing coal and gas power capacity can be retrofitted for hydrogen co-firing, reducing stranded capacity, mainly in the Organization for Economic Co-operation and Development (OECD) countries and Asia. However, electricity supply from co-firing generators is limited to about 1% of total electricity generation, because hydrogen co-firing is mainly used as a backup option to balance the variable renewable energies. The incremental fuel cost of hydrogen results in lower capacity factor of hydrogen co-fired generators, whereas low-carbon hydrogen contributes to reducing emission cost associated with carbon pricing. While hydrogen co-firing may play a role in balancing intermittency of variable renewable energies, it will not seriously delay the phase-out of fossil-based generators.

The Paris Agreement set climate goals of a temperature increase well below 2 °C and to pursue efforts for below 1.5 °C. The corresponding mitigation pathways require huge and rapid reduction of $CO_2$ emissions from the energy sector; in particular, the power sector needs to be decarbonized rapidly in the coming decades[1–3]. Given the recent trends and future expectation of decreasing costs for solar photovoltaic (PV) and wind power[4,5], mitigation strategies proposed for power system decarbonization generally involve upscaling of low-carbon electricity, especially renewable sources replacing fossil fuel-fired generators[6,7]. Such mitigation scenarios generally involve phase-out and retirement of fossil fuel-fired powerplants before their expected lifetime, resulting in stranded assets[8–10]. Although carbon capture and storage (CCS) would facilitate fossil fuel-fired power generation under the low emissions scenarios, CCS-based electricity generation is limited in the mitigation scenarios of the Intergovernmental

Panel on Climate Change Sixth Assessment Report (IPCC AR6), with a median value <10% in 2050[11] due to several barriers, such as incremental capital costs and energy penalties[12,13].

Hydrogen is expected to contribute to deep decarbonization, replacing fossil fuels in sectors that are hard to decarbonize, such as long-distance transport and some industry sectors[14–16]. Meanwhile for the power sector, flexible generators that integrate variable renewable energies (VREs), such as gas-fired generators, will be difficult to decarbonize[17]. However, previous studies have not intensively focused on application of hydrogen to power sector decarbonization. Hydrogen co-firing, including hydrogen-based energy carriers such as ammonia, could be an option of $CO_2$ emissions abatement from fossil fuel-fired generators as long as the necessary hydrogen is generated from low-carbon sources, such as renewable electricity and biomass[18,19]. In addition, hydrogen could potentially contribute to VRE

[1]Kyoto University, C1-3, Kyotodaigaku-Katsura, Nishikyo-ku, Kyoto, Japan. [2]National Institute for Environmental Studies, Tsukuba, Japan. [3]International Institute for Applied Systems Analysis (IIASA), Laxenburg, Austria. ✉e-mail: ohshiro.ken.6e@kyoto-u.ac.jp

integration as a seasonal storage option for electricity[20,21]. Given the recent decreases in costs of solar PV and wind power generators and the expectation of increasing low-carbon hydrogen supply through electrolysis[4,5], hydrogen co-firing may be an option for power system decarbonization.

Hydrogen is a secondary energy carrier that is obtained from various energy sources. Therefore, its role has been assessed using energy system models and integrated assessment models (IAMs), which cover both energy supply and demand sectors, as well as power system models. Nevertheless, as only a few studies have assessed hydrogen use in the power sector, there are still several knowledge gaps associated with the potential role of hydrogen. First, previous studies have mainly focused on hydrogen co-firing only with natural gas-fired generators and with a limited regional coverage. Öberg et al.[18] used a power system model for European countries and found that hydrogen co-firing is limited benefit in natural gas powerplant, even in a stringent mitigation scenario. Bui et al.[22] evaluated the $CO_2$ emission intensity of hydrogen-based generators and indicated that it can contribute to decarbonizing natural gas generators such as CCS. In national model intercomparison studies for Europe and the US, some models have included hydrogen-based energy carriers, concluding that hydrogen is mainly used in high-density transport fuels and high-temperature industrial processes, whereas its use in the power sector is limited[23,24]. Although these studies have provided insights into the role of hydrogen, the potential of hydrogen co-firing at the global level, including at coal and at gas powerplants, is not yet well understood. Second, although retrofitting fossil-fueled generators with hydrogen co-firing can theoretically reduce stranded asset risks, little is known about such effects. Because global coal power generation is currently still rising, especially in Asian countries[25], stranded asset risk can be a significant issue in these regions[10,26,27]. Although previous studies have assessed the effect of retrofitting coal powerplant with carbon capture and biomass co-firing[28,29], the potential impact of hydrogen co-firing on avoiding stranded asset risks has not yet to be explored.

Against these backgrounds, the question of whether hydrogen co-firing could reduce the transition risks associated with the phase-out of fossil-fueled generators and stranded assets arises. As described in the International Energy Agency (IEA) Net-Zero report[30], hydrogen co-firing could theoretically lead to continued use of existing coal- and natural gas-based assets under the energy system transition scenario. The Group of Seven (G7) Climate, Energy and Environment Ministers' Communiqué of April 2023[31], acknowledges the roles of hydrogen and ammonia in the power sector towards net-zero emissions, although their impacts have not been explored. To clarify this research question, we develop a global energy system model with high temporal resolution with a variety of technology options, including hydrogen co-firing. In this study, we explore the potential role of hydrogen co-firing on decarbonization of the global energy system.

Quantitative scenario assessment is conducted using the Asia-Pacific Integrated Model (AIM)-Technology, which is a global bottom-up energy system model. Although hydrogen production and use in the final energy sectors have been modeled in previous studies[14,32], there are three major advancements associated with hydrogen applications in the power sector to address the research question of this study. First, gas- and coal-fired power generators with hydrogen co-firing are represented to assess their roles in global mitigation scenarios. It is assumed that both hydrogen and ammonia can be co-fired for these generators (hereafter, hydrogen co-firing is defined as including both hydrogen and ammonia). Since the operating conditions of co-fired generators are determined endogenously in the model similar to other fossil fuel-fired generators, hydrogen co-fired generators can play a dual role in complementing the intermittency of VRE and serving as base or middle-load generators. Additional investments required for hydrogen co-fired generators, which vary with the hydrogen mix rate, are assumed based on the literature[18].

Second, the model is updated to represent the retrofitting of existing fossil fuel-fired generators with hydrogen co-firing and carbon capture technologies. Consequently, a technology retrofit is endogenously determined in the model, based on the additional investment required for the retrofit, the changes in energy and emissions performances, and the remaining lifetime of the upgraded plant. To quantify the amount of stranded assets in the mitigation scenarios, existing and planned coal and gas powerplant information is obtained from the literatures[33,34]. Third, to consider the seasonal variation in electricity demand and supply from VREs, the intra-annual temporal resolution is improved in this model. As previous studies assessing seasonal storage modeled monthly power supply and demand profiles[35,36], the AIM-Technology in this study also contains 12 representative days corresponding to one per month and 24 h per day. More details about these changes are presented in the Methods section.

Multiple scenarios are prepared to assess the outcomes of hydrogen co-firing in various mitigation pathways and technology portfolios. The emissions pathways are based on carbon budgets by 2100 of 500, 700, 1000 and 1400 gigatonnes (Gt) -$CO_2$[37]. In the 500 Gt-$CO_2$ scenario, which is compatible with the 1.5 °C goal, energy-related $CO_2$ emissions are reduced to nearly net zero by 2050[14]. Scenarios are also classified based on technology availability to explore conditions of hydrogen co-firing. The Default scenario is based on the default model setting with no any additional constraints or parameter changes. The H2-optimistic (H2Opt) scenario considers the drastic cost reductions for electrolyzer and solar and wind power, which are expected to enhance hydrogen co-firing. The Limited-CCS (LimCCS) scenario, where geological storage and bioenergy supply are constrained to 4 Gt-$CO_2$ per year and 100 exajoule (EJ) per year, respectively, is assessed because it requires stringent residual emissions reductions and associated increases of carbon prices[14]. The H2Opt+ scenario is a what-if scenario with the most optimistic conditions for hydrogen co-firing. In addition to the conditions in the H2Opt and LimCCS cases, the H2Opt+ scenario includes higher cost assumptions for battery storage and no new construction of seasonal storage. The No-Cofire (NoCOF) scenario is assessed as a reference wherein hydrogen co-firing with fossil fuel-fired powerplants is not available. Sensitivity scenarios are also analyzed for various socio-economic conditions. More details about the assumptions of these scenarios are provided in the Methods.

## Results
### Hydrogen consumption in the power sector
Upscaling of low-carbon electricity, particularly solar and wind power, is commonly observed across all mitigation scenarios and regions (Fig. 1a, Supplementary Fig. 1). Consequently, $CO_2$ emissions from energy supply sectors fall rapidly and range around −5 to 5 Gt−$CO_2$ in 2050 under all the mitigation scenarios, while residual emissions from energy demand sectors account for around 5−15 Gt−$CO_2$ in 2050 (Supplementary Fig. 2a, Supplementary Fig. 3). Carbon prices range from around 150 US $ per t-$CO_2$ or higher by 2050, reaching 300−600 US $ per t-$CO_2$ in the 500 Gt−$CO_2$ scenarios, where energy-related $CO_2$ emissions decreases to nearly net zero in 2050 (Supplementary Fig. 2b). Cumulative mitigation cost increases with increasing mitigation stringency, ranging around 0.9−1.1% of gross domestic product (GDP) in the 500 Gt−$CO_2$ scenarios (Supplementary Fig. 4).

A phase-out of fossil fuels in the power sector is generally observed across all mitigation scenarios, even when hydrogen co-firing is available. Power generation from fossil fuel-fired generators, which includes hydrogen co-firing, decreases rapidly, accounting for around <20 EJ per year by 2050 (Fig. 1b). These values are similar to the ranges obtained from the corresponding mitigation scenarios of the IPCC-AR6. A trend of decreasing fossil fuel use is also observed in the total primary energy supply across all regions (Supplementary Fig. 5). Although total power generation from fossil fuel-based generators decreases, stringent mitigation scenarios may employ hydrogen co-firing. In the 500 Gt−$CO_2$

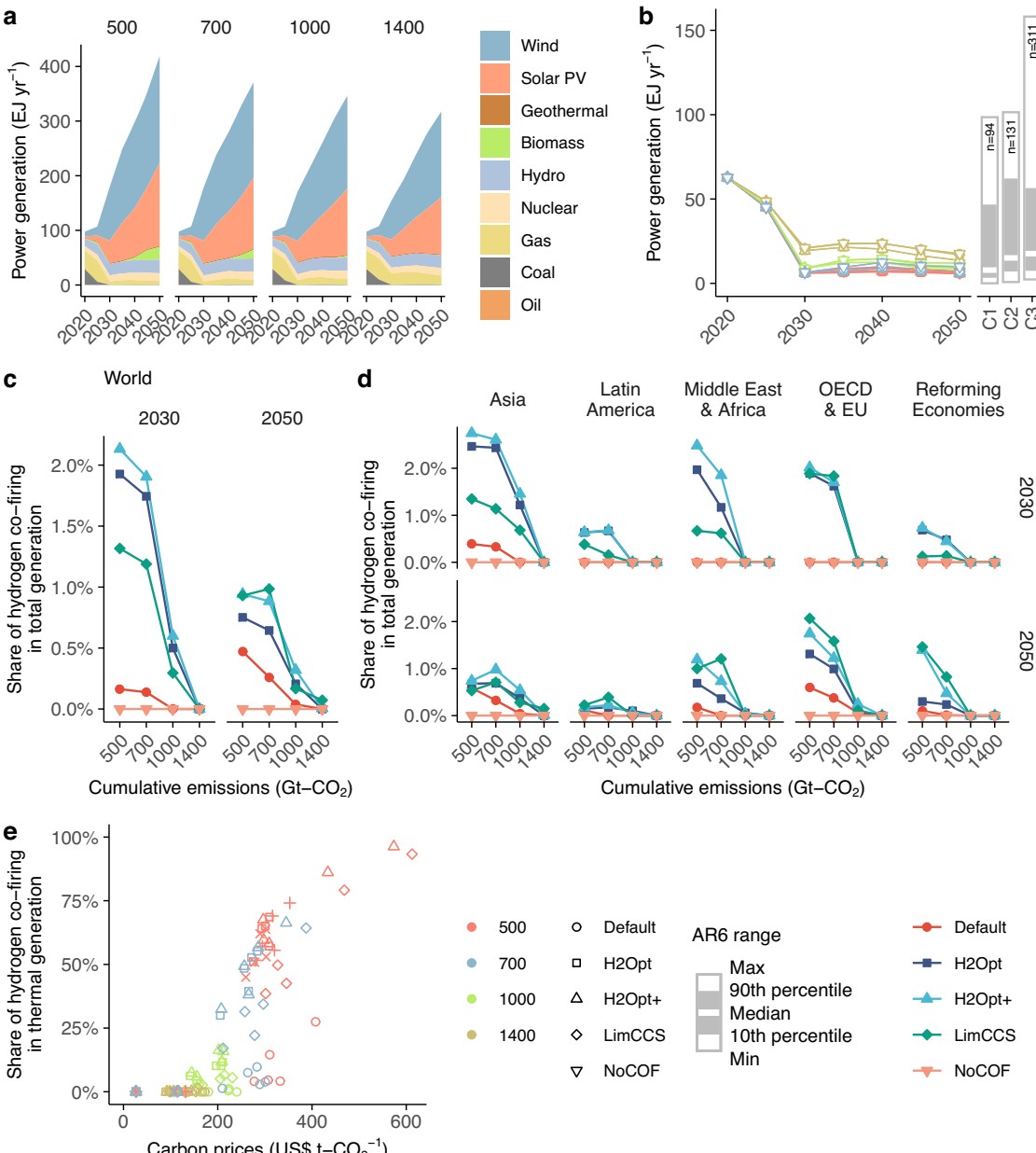

**Fig. 1 | Contribution of hydrogen co-firing to power generation. a** Power generation in the 500, 700, 1000 and 1400 Gt–CO₂ scenarios with Default technology. Results for other scenarios are shown in Supplementary Fig. 1. **b** Power generation from fossil fuel-fired generators (including hydrogen co-firing and carbon capture and storage [CCS]). Right bar plots illustrate the power generation from fossil fuels in 2050 as obtained from the Intergovernmental Panel on Climate Change Sixth Assessment Report (IPCC AR6) Scenario Database[82] for each climate category (C1–C3). "*n*" denotes the number of available scenarios in each category. **c**, **d** Hydrogen co-firing as a share of total power generation in 2030 and 2050 in the world, and Asia, Latin America, Middle East and Africa, the Organization for Economic Co-operation and Development (OECD) and Europa (EU), and Reforming Economies, respectively. Results for other scenarios are shown in Supplementary Fig. 7. **e** Hydrogen co-firing as a share of fossil fuel-based power generation relative to carbon prices, excluding No-Cofire (NoCOF) cases.

scenarios excluding the NoCOF scenarios, power generation from hydrogen co-fired generators reaches around 2–6 EJ per year in 2050, equivalent to around 30–90% of total fossil fuel-fired power generation (Supplementary Fig. 6). Nevertheless, the share of power generation associated with hydrogen co-firing is negligible compared to total power generation, accounting for <1% in 2050, including that for sensitivity scenarios with various technological and socio-economic conditions (Fig. 1c). Furthermore, in scenario with carbon budgets >1000 Gt–CO₂, diffusion of hydrogen co-firing is very low, even for H2Opt+. In 2030, hydrogen co-firing is observed in the LimCCS, H2Opt, and H2Opt+ cases reaching ~2–4 EJ per year in the stringent mitigation scenarios, while hydrogen co-firing implementation is absent or very

limited in the Default cases. Relative to the global average, the share of hydrogen co-firing in total power generation rises in the Organization for Economic Co-operation and Development (OECD) and Asian countries, followed by the Middle East and Africa; however, the contributions of hydrogen co-firing to total power generation remain limited, reaching 2–3% in each region (Fig. 1d).

Mitigation stringency is a driver of hydrogen co-firing, as this process is implemented when carbon prices exceed about 200 US $ per t-CO₂ in the H2Opt, H2Opt+ and LimCCS cases, whereas it occurs in the Default scenarios at the carbon prices >300 US $ per t-CO₂ (Fig. 1e). As hydrogen production depends on low-carbon electricity in these scenarios (Supplementary Fig. 8), hydrogen co-firing is an effective

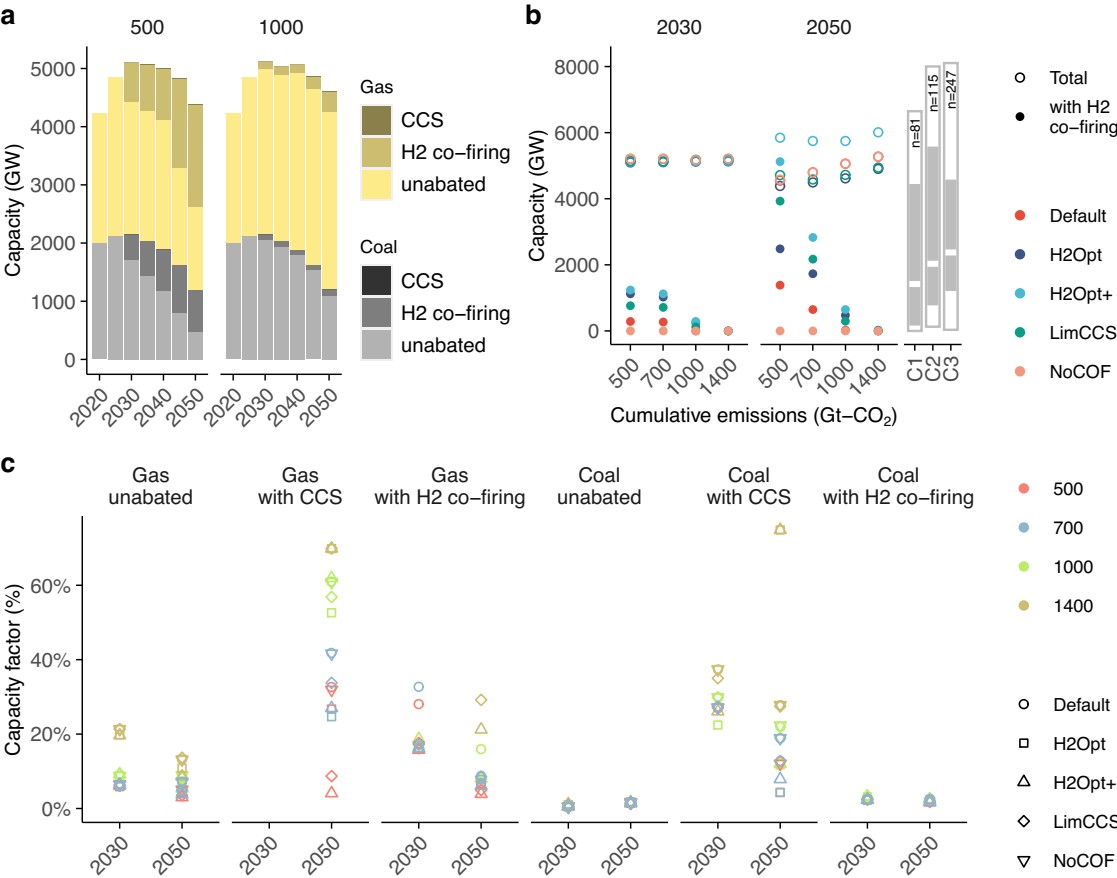

**Fig. 2 | Capacity and capacity factor of hydrogen co-fired generators. a** Capacity of electricity generators from fossil fuels including hydrogen co-firing in the 500 and 1000 Gt–CO$_2$ scenarios with H2-Optimistic (H2Opt) conditions. Results for other scenarios are shown in Supplementary Fig. 12. **b** Capacities of total fossil fuel-fired generators and hydrogen co-firing in 2030 and 2050 across various climate and technology scenarios. Right bar plots illustrate the capacity of total fossil fuel-fired generators in 2050 in the Intergovernmental Panel on Climate Change Sixth Assessment Report (IPCC AR6). "*n*" denotes the number of available scenarios in each category. **c** Annual average capacity factors of coal- and gas-fired generators in 2030 and 2050.

method to reduce CO$_2$ emissions from energy supply sectors at such high carbon prices. Consequently, hydrogen co-firing of fossil fuel-based generators contributes to reducing CO$_2$ emissions intensity, accounting for around 0.075 t-CO$_2$ per GJ in the 500 H2Opt scenario and 0.1 t-CO$_2$ per GJ in the 500 NoCOF scenario (Supplementary Fig. 9).

In addition to hydrogen utilization in the power sector, hydrogen and hydrogen-based energy carriers, including ammonia and synthetic hydrocarbons, also help reduce emissions in the final energy demand sectors (Supplementary Fig. 10). The main consumer of hydrogen is the transport sector, followed by other energy demand sectors (Supplementary Fig. 11). In comparison, hydrogen consumption in the power sector is relatively small.

**Capacity factor of hydrogen co-fired generators**

Although the impact of hydrogen co-firing on total electricity generation is limited, the capacity of hydrogen co-fired generators scales up rapidly, especially under stringent mitigation scenarios (Fig. 2a). In particular, in the 500 H2Opt, 500 H2Opt+ and 500-LimCCS scenarios, the capacity of hydrogen co-fired generators increases to more than half of total fossil fuel-fired powerplants in 2050, reaching about 2000, 5000, and 4000 gigawatts (GW), respectively (Fig. 2b). In these scenarios, around half of capacity for both coal- and gas-fired generators is equipped with hydrogen co-firing technology. It should be noted that the power capacity values in this study are larger than those in the IPCC-AR6 scenarios, as shown in Fig. 2b, because the AIM-Technology model has detailed time resolution, which results in a large requirement for back-up capacity for VRE intermittency. Despite the increase

in hydrogen co-firing capacity, the annual average capacity factor of hydrogen co-fired powerplant is much lower than under today's conditions, accounting for <30% and 5% of gas and coal generators, respectively (Fig. 2c). This difference arises because fossil fuel-fired generators with hydrogen co-firing are used as a backup source for balancing VRE intermittency, especially under stringent mitigation scenarios, resulting in these generators operating for only a few hours per year (Supplementary Fig. 13). In seasons when power generation from VRE is abundant, direct use of electricity from VREs is a more efficient option rather than converting the electricity into hydrogen due to conversion losses in the electrolysis and hydrogen-to-power conversion processes. In contrast, fossil fuel-fired generators with CCS have a higher capacity factor than hydrogen co-firing in 2050, reaching up to 70% and 30% for gas and coal, respectively. Nevertheless, the contribution of CCS plants to total power generation remains limited due to their cost penalty.

**Role of hydrogen in seasonal storage**

While the power supply from hydrogen co-fired generators is limited in terms of total annual power generation, its contribution to balancing the seasonal variability of electricity supply is notable, especially under stringent mitigation scenarios (Supplementary Figs. 13, 14). Figure 3a compares the annual power supply provided by various seasonal storage options. While pumped hydro energy storage (PHES) for seasonal balancing (S-PHES) and compressed air energy storage (CAES) are commonly used and effective seasonal storage options across mitigation scenarios in 2050, the contribution of hydrogen co-firing varies

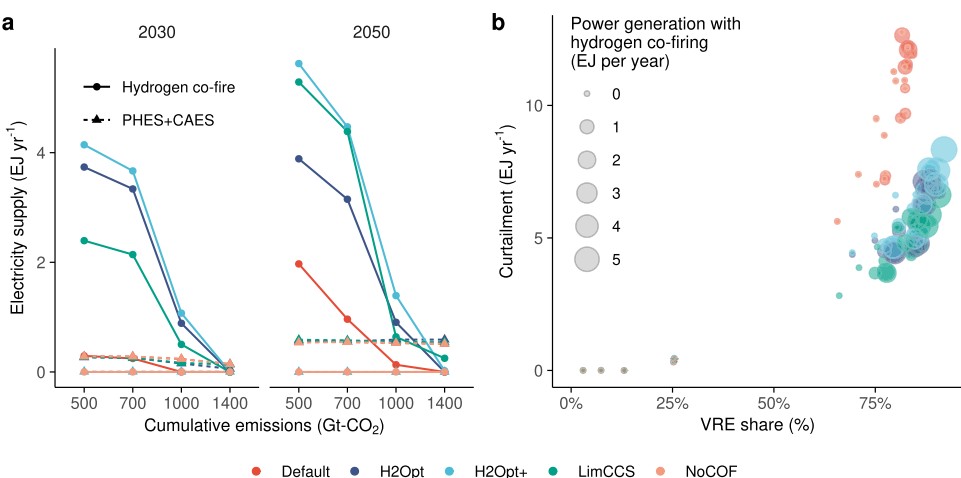

**Fig. 3 | Role of hydrogen utilization to offset variable renewable energy (VRE) intermittency in the power sector. a** Electricity supply from seasonal storage technologies. Dotted line denotes those from pumped-hydro electricity storage (PHES) and compressed air electricity storage (CAES). **b** Curtailment of generated electricity relative to the share of VREs in power generation. Bubble size represents the amount of power generation with hydrogen co-firing.

with the stringency of mitigation as well as technological conditions associated with hydrogen. In 2050, the collective power supply from S-PHES and CAES reaches around 0.5 EJ per year with little difference among scenarios, as these technologies are cost-effective options for addressing VRE integration despite their limited potential[38]. In contrast, the role of hydrogen co-fired generators on seasonal power storage is limited under 1000 Gt−$CO_2$ and higher scenario, even in the H2Opt+ cases. Nevertheless, the power supply from hydrogen co-firing exceeds those of S-PHES and CAES in the 500 and 700 Gt−$CO_2$ scenarios. Hydrogen co-firing also has an impact on battery storage capacity with respect to short-term variability (Supplementary Fig. 15); however, it would not reduce the importance of battery as a short-term storage option, because the difference in battery capacity between the default and NoCOF scenarios is relatively small.

The role of hydrogen co-firing as seasonal storage is highlighted by the reduced curtailment of excess VRE supply (Fig. 3b). While curtailment of electricity increases across all scenarios as the VRE share of power generation increases, reaching ~10 EJ per year, the amount curtailment is almost halved under the H2Opt, H2Opt+, and LimCCS cases relative to the Default and NoCOF cases, at 3–7 EJ per year.

**Impacts on stranded capacity of fossil fuel-fired generators**
As the mitigation scenarios require rapid emissions reductions in the near-term, stranded coal capacity in 2030 reaches 1500 GW in the NoCOF cases, equivalent to about two thirds of existing coal capacity (Fig. 4a, b), which is roughly comparable with previous reported values[26]. Stranded capacity is observed mainly in Asia, especially for coal-fired generators, followed by OECD countries, the Middle East and Africa (Fig. 4c), due to the near-term greater capacity of coal- and gas-fired generators (Supplementary Fig. 18). In 2050, stranded coal capacity falls to ~500 GW, accounting for around one third of total existing capacity, due to the retirement of existing capacity. Stranded capacity is lower for gas-fired generators than coal-fired plants, accounting for around 500 GW and 100 GW in 2030 and 2050, respectively.

Hydrogen co-firing can reduce the amount of stranded capacity of coal- and gas-fired generators under some mitigation scenarios, but is not always cost-effective. Stranded coal capacity is reduced by about 200 GW in the H2Opt and H2Opt+ cases, meaning that it is nearly halved in 2050, whereas the impact on gas-fired generators is much smaller. In contrast, stranded coal and gas capacities in the Default and LimCCS cases do not discernably differ from the NoCOF case. This pattern emerges because fossil fuel-fired generators without co-firing,

especially gas-fired powerplants, are an effective option for backup power to support VRE integration[39]. Although hydrogen co-firing can replace this backup capacity and thereby reduce associated emissions, its impact on stranded capacity is limited.

**Cost implications of hydrogen application in the power sector**
The limited potential of hydrogen co-firing in the global power systems is mainly due to the cost penalty of hydrogen relative to other low-carbon generators. Figure 5a presents a comparison of the levelized cost of electricity (LCOE) across major power generators while considering the effect of carbon prices. Hydrogen co-fired generators generally have much higher LCOE than generators without co-firing under moderate mitigation scenarios because hydrogen is more costly to produce than fossil fuels, at around 30–40 US$ per GJ (Supplementary Fig. 19). Differences in LCOE between generators with and without co-firing narrow under stringent mitigation scenarios. In particular, in the H2Opt and H2Opt+ cases, the LCOE of hydrogen co-fired generators is similar to or lower than the value for generators without co-firing.

Such variation in LCOE among scenarios derive mainly from heterogeneity in fuel and emissions cost compositions (Fig. 5b). While hydrogen co-firing is generally associated with increases fuel costs and capital costs under the most mitigation scenarios, these changes can be offset by reduced emissions costs associated with higher carbon prices under stringent mitigation scenarios. Specifically, for the H2Opt and H2Opt+ cases, LCOE of hydrogen co-fired generators is similar to that without co-firing due to the fuel cost reduction relative to Default cases. Nevertheless, hydrogen co-firing is generally more costly than other low-carbon generators, such as solar PV, even if the cost of hydrogen falls substantially. Therefore, the contribution of hydrogen co-fired generators to total power generation is limited, and they are employed only as backup capacity for VREs due to their high variable costs[40].

## Discussion
According to the scenario assessment with various assumptions about mitigation stringency and the technology portfolio, the impact of hydrogen co-firing on prolonging fossil fuel-based power generation is limited. Even in the H2Opt+ scenario in this study, which was designed as a what-if scenario that assumes the most optimistic condition for hydrogen co-firing, the share of hydrogen co-firing accounts for <1% of global total power generation in 2050. The hydrogen co-firing results differ among regions; the share of hydrogen co-firing is larger in OECD

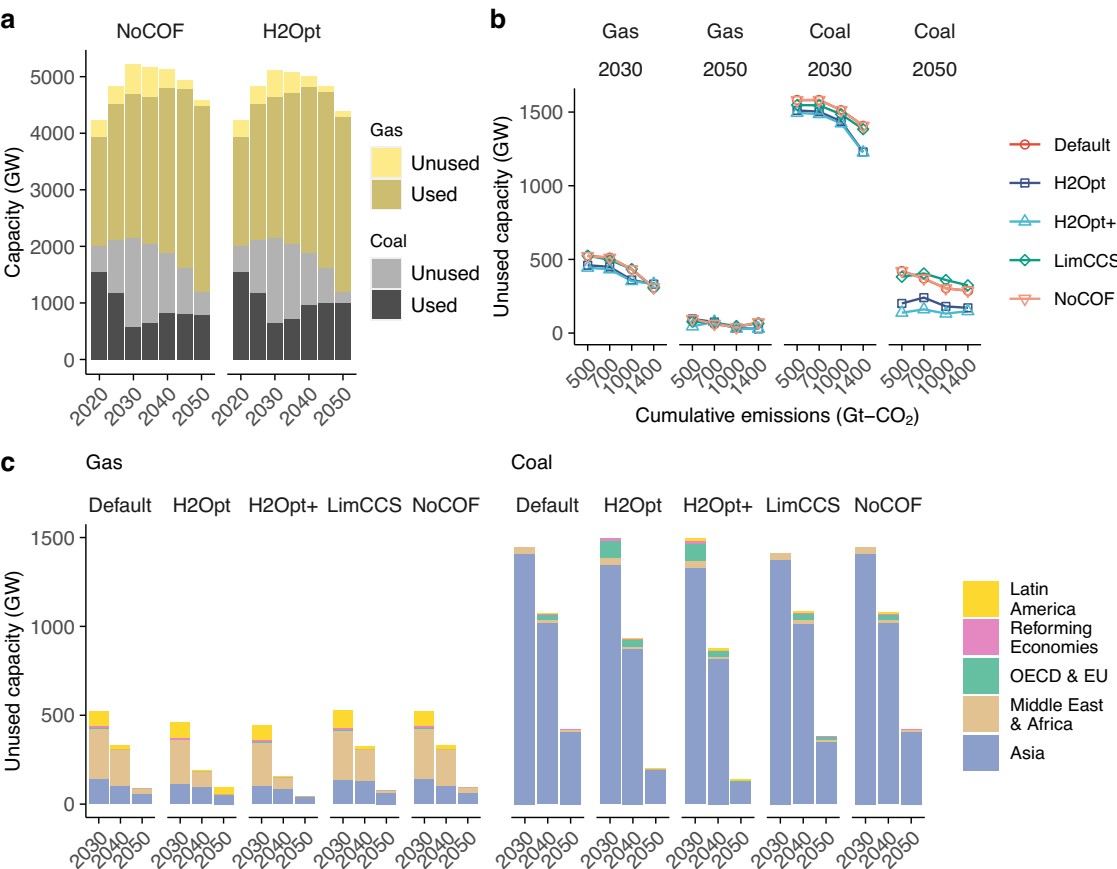

**Fig. 4 | Impacts of hydrogen co-firing on stranded capacity of power generators. a** Capacity of fossil fuel-fired generators in the 500 Gt–CO$_2$ scenarios. Results for other scenarios are shown in Supplementary Fig. 16. **b** Unused power capacities of coal- and gas-fired generators. **c** Unused coal and gas power capacities by region for the 500 Gt-CO$_2$ scenarios in Asia, Latin America, Middle East and Africa, the Organization for Economic Co-operation and Development (OECD) and Europa (EU), and Reforming Economies. Results for other scenarios are shown in Supplementary Fig. 17.

countries and Asia because of the greater risks of stranded capacity. Nevertheless, the impact of hydrogen co-firing is limited even in these regions, as the share of hydrogen co-firing reaches 2–3% at most. Although expansion of hydrogen co-fired generator capacity is observed for both coal and gas, the contributions of these changes to total power generation are small, because the hydrogen co-fired generators mainly used as a back-up option for VRE intermittency. It is due to greater cost penalty associated with hydrogen production, relative to the direct use of generated electricity from renewables or energy penalty of CCS implementation. Although hydrogen cost reduction may enhance hydrogen co-firing, the phase-out of fossil fuel-fired generators is a robust trend in the deep decarbonization scenario. These results suggest that the transition risks of fossil fuel-based power infrastructure must be considered when developing climate and energy strategies, even when hydrogen co-firing becomes technically feasible.

Despite the limited impacts on total power generation, the results of this study clarify the potential roles of hydrogen in deep decarbonization. While previous studies have mainly highlighted the role of hydrogen utilization in energy demand sectors that are difficult to decarbonize[14,15,23], such as long-distance transport and high-temperature industrial processes, this study demonstrates that low-carbon hydrogen can contribute to power sector decarbonization specifically in the form of flexible generators to account for the seasonal intermittency of VREs. Nevertheless, our results underscore that hydrogen co-fired generators have a minor influence on the power generation mix. This is attributed to the presence of other cost-effective options for integrating VRE, including PHES

and CAES, battery storage, and curtailment. Because seasonal electricity storage with hydrogen is a relatively expensive option for VRE integration, the development and deployment of various technological options, rather than depending solely on hydrogen, will be essential.

Although we confirmed the potential roles of hydrogen co-firing in power system decarbonization, an absence of hydrogen co-firing would have no substantial impact, as net zero emissions are accomplished in the NoCOF scenarios without extensive mitigation cost increases. In contrast, limited availability of CCS and biomass substantially increases mitigation costs, as shown in the LimCCS cases. This difference occurs because alternative decarbonization options to hydrogen co-firing exist for the power sector, such as the direct use of VREs and CCS. In contrast to the power sector, hydrogen use in the hard-to-decarbonize sectors of transport and industry could play an essential role[14]. In this regard, energy strategies to support hydrogen penetration should be based on a holistic approach that prioritizes the effective use of hydrogen.

Some limitation of caveats on the interpretation of our results should be noted. First, as national energy strategies generally involve ensuring energy security as well as economic and environmental aims, countries still maintain fossil fuel reserves today. Because hydrogen and hydrogen-based carriers are relatively easier to store than other low-carbon sources, such as solar- and wind-based electricity, hydrogen use in the power sector can be a practical option for energy security purposes. Second, as this study is based on single model assessment, it would be useful to employ other models with different structures, such as different temporal resolutions. In

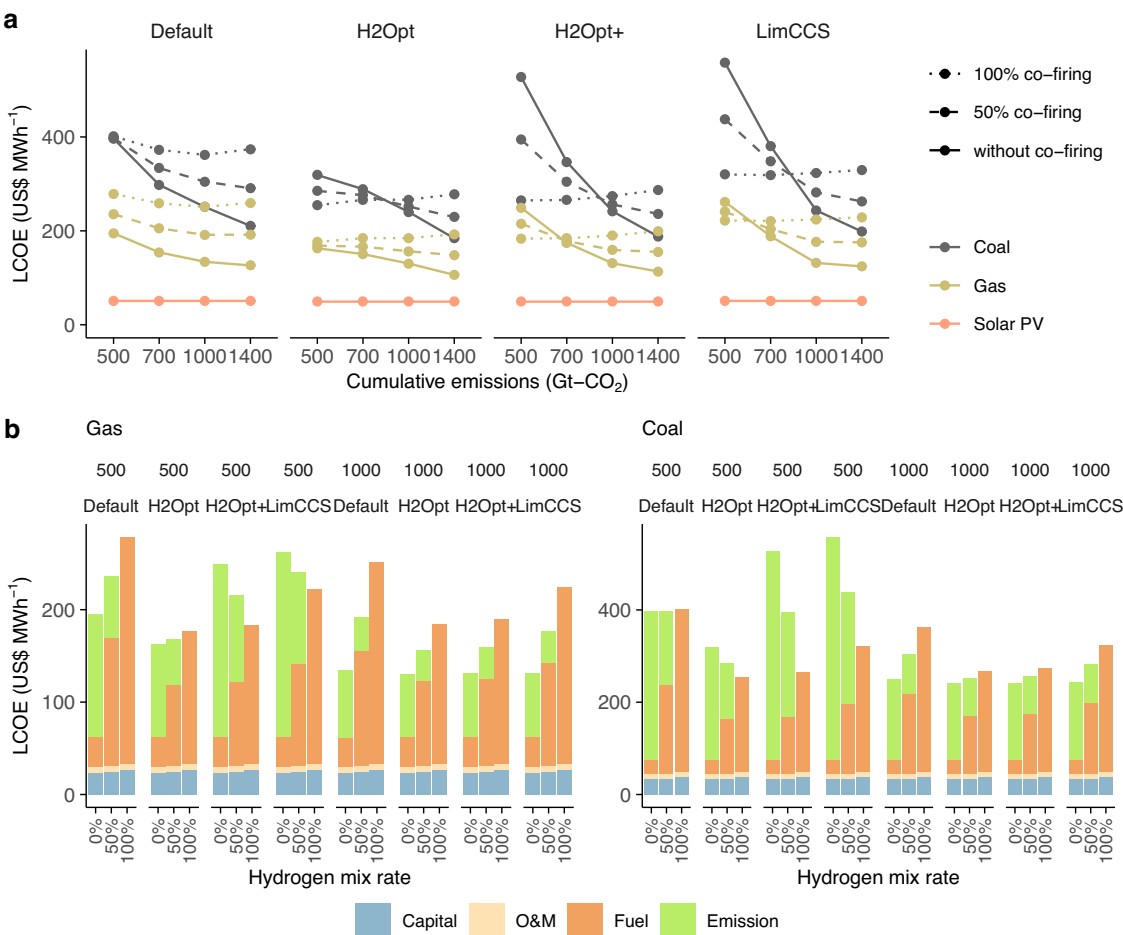

**Fig. 5 | Cost implications of hydrogen co-firing. a** Levelized cost of electricity (LCOE) comparison among different power generation technologies in 2050 with a 10% discount rate, including the effect of the carbon price penalty on $CO_2$ emissions. Integration cost is not included. **b** LCOE decomposition in 2050 for the 500 and 1000 Gt–$CO_2$ scenarios. Results for other scenarios are shown in Supplementary Fig. 20.

addition, although this study indicates that extensive wind power generation exceed the power generated by solar PV in 2050, which is consistent with recent scenario assessments using the Global Change Analysis Model (GCAM) and Model for Energy Supply Strategy Alternatives and their General Environmental Impact (MESSAGEix)[41,42], expectations of increased solar PV are growing given recent developments. Although the model choice and structure can affect the quantitative result of power generation to some extent, such as the requirement for seasonal storage and solar PV expansion, the findings of a limited role of hydrogen co-firing on decarbonization is likely to be robust, because the cost disadvantage of hydrogen co-firing would not be greatly affected by the choice of model. Third, solar- and wind-based electricity systems would be more vulnerable to extreme conditions, such as extreme weather events and natural disasters. Under such extremes conditions, a backup power supply provided by hydrogen is an attractive option.

## Methods

### Energy system model

Quantitative analysis of scenarios was conducted using AIM-Technology which is a global bottom-up energy system model[14,32]. In this model, the operating conditions of several energy technologies, including both energy supply and demand sectors, are determined through linear programming to minimize total energy system cost. The total energy system costs include the annualized initial costs of technologies, energy, operating and emissions costs, which are subject to exogenous energy service demand. Technological

parameters including energy efficiency and cost parameters, energy service demands, and several constraints such as primary energy resources are provided to the model as exogenous parameters. Quantitative information about energy demand, energy supply, $CO_2$ emissions and sequestration, energy system costs, and carbon prices are calculated and output from the model. In the energy supply sector, several primary and secondary energy sources, including fossil fuels, nuclear, renewables and hydrogen, are converted to electricity. The power sector uses a dispatch module with a 1 h temporal resolution for representative days. The operating conditions of the power generator are determined under a cost-minimizing framework. Therefore, the capacity factor of flexible generators, including hydrogen co-fired generators, is determined endogenously in this model. In terms of hydrogen generation, hydrogen production from fossil fuels, biomass, and electrolysis is modeled. The mathematical equations, parameter settings and further detailed information have been reported in Oshiro et al.[14].

In this study, gas- and coal-fired power generators employing hydrogen co-firing were added to the model. Incremental investment in hydrogen co-firing for both new construction and retrofitting was assumed based on an existing study[18] as summarized in Supplementary Table 1. Because a higher hydrogen mix rate requires additional investment for upgrading of the combustion chamber as well as hydrogen tank, incremental costs of hydrogen co-firing varied with hydrogen mix rate. We assumed that both hydrogen and ammonia could be co-fired in both coal- and gas-fired generators. In the context of cost minimization, the hydrogen mix rate was internally determined

in the model within 0–100%, with a 100% mix indicating exclusive hydrogen combustion.

In addition, the model was updated to represent retrofitting of existing fossil fuel-fired generators with hydrogen co-firing and carbon capture. In the updated model, a technology retrofit is endogenously determined based on the incremental cost for the retrofit, as summarized in Supplementary Table 1, and on a cost minimizing framework. The annualized incremental cost is calculated based on the remaining lifetime of the upgraded plant. The mathematical equations for technology retrofitting are summarized in the Supplementary Note 1. To quantify stranded assets in the mitigation scenarios, information about existing and planned coal and gas powerplant was obtained from the Global Coal Plant Tracker[33] and the Global Gas Plant Tracker[34]. Planned powerplants included those under construction or in the permitting processes. To incorporate recent trends in solar PV and wind power capacity expansion, the calibration period for solar and wind capacity has been extended to 2027, based on the IEA's estimates[43].

To account for the seasonal variation in electricity demand and supply from VREs, the intra-annual temporal resolution was improved in this model version to include 12 representative days, corresponding to one for each month, and 24 h per day, whereas the previous model used two typical days to represent year, corresponding to summer and winter[14]. Due to tradeoffs between the model temporal resolution and computational resource requirements[44], there have been few presentations of the intra-annual temporal resolution in the IAMs. Recently, some IAMs have modeled the seasonal specific electricity supply and demand profiles explicitly[39,45], or soft-linked the IAMs and detailed power system models[46–48]. By contrast, detailed power and energy system models, such as The Integrated MARKAL-EFOM System (TIMES), present a more detailed temporal resolution for four seasons or for each month[49–51]. Although there are several methods to reduce the time resolution for power system analysis[52,53], the representative day approach for each season or month offers a clear advantage in assessing seasonal storage effects, as it represents seasonal specific supply and demand conditions. Because the 2–3-season average approach is insufficient to capture the duration curves of solar and wind[54], and several studies assessing electricity storage modeled monthly power supply and demand profiles[35,36,55], the updated model contains 12 representative days, one for each month, and 24 h per day. In addition, four representative days, when the power output from VRE generators is at its lowest in each quarter of the year, were selected based on historical weather observations between 2006 and 2015, obtained from of the Modern-Era Retrospective analysis for Research and Applications Version 2 (MERRA-2)[56,57] to consider the impacts of VRE intermittency due to weather conditions. For each representative day, typical regional monthly electricity demand profiles were determined from the literature[58,59]. VRE potential and hourly output in each month were also estimated based on the literature[60,61]. PHES and CAES were introduced as new technologies that can compete with hydrogen co-firing as seasonal storage options. Their technological potentials were assumed based on the literature[62,63]. Technological parameter assumptions for these technologies are summarized in Supplementary Table 2.

There are various methods for model evaluation of process-based IAMs[64]. Model intercomparison has been conducted with a number of IAMs[1,65], and some of the models have included model documentation[66,67]. In some studies, historical simulations[68,69] and diagnostic studies[70,71] have been conducted. For the AIM-Technology model, detailed model documentation, including mathematical equations and parameter assumptions, is available[72]. Model intercomparisons have been conducted for national studies[73–75]. A recent study using the AIM-Technology-Global model presented a comparison with the IPCC-AR6 scenario[32].

Based on the simulation results of energy system models, several indicators were calculated to clarify the potential role of hydrogen co-firing. First, the stranded capacity of powerplants was estimated as unused capacity through a target year, in accordance with previous research[76]. Second, as a cost indicator for comparison among power generation technologies, LCOE was calculated based on a 10% discount rate.

## Scenarios

Multiple scenarios were modeled to assess the condition of hydrogen co-firing under various mitigation pathways and technology portfolios (Supplementary Table 3). The emissions pathways used in this study were based on carbon budgets by 2100 of 500, 700, 1000 and 1400 Gt–$CO_2$[37]. The 500 Gt–$CO_2$ scenario corresponds to the 1.5 °C goal of the Paris Agreement, and energy-related $CO_2$ emissions reach nearly zero in 2050. Emissions reduction begins from 2024 in all mitigation scenarios, aligning with the historical emissions trend up to 2023[77].

Scenarios were also classified based on technology availability and assumptions to explore the potential of hydrogen co-firing. The Default case was based on the model's default settings, in which hydrogen co-firing is available, and no additional constraint or parameter changes were imposed. The H2Opt case considered a dramatic cost reduction for hydrogen electrolyzer and solar and wind power, which could enhance hydrogen co-firing by lowering the production cost of hydrogen. The costs of solar and wind power in the H2Opt case were based on the International Renewable Energy Agency (IRENA) estimates[78,79]. Electrolyzer cost in the H2Opt case was based on the IEA Net-Zero report[30]. The cost assumptions on these technologies are summarized in Supplementary Fig. 21. Furthermore, additional technological conditions were included to elucidate the possible role of hydrogen utilization in the power sector. In addition, because limited CCS and bioenergy could lead to stringent reductions of residual emissions and associated increases in carbon prices[14], the LimCCS scenario was prepared. In this scenario, geological storage of captured $CO_2$ and bioenergy supply are limited to 4 Gt-$CO_2$ per year and 100 EJ per year, respectively, based on the literature[6]. In the LimCCS case, electrolyzer and renewable energy costs are the same as in the H2Opt case. The H2Opt+ scenario was designed as a what-if scenario assuming the most optimistic conditions for hydrogen co-firing. In addition to the conditions imposed in the H2Opt case, this scenario includes higher battery cost assumptions and no new construction of seasonal storage (CAES an PHES). Also, the same limitations on CCS and bioenergy that occur in the LimCCS case were imposed in the H2Opt+ case. The NoCOF scenario was assessed as a reference scenario for comparison with other scenarios, wherein hydrogen co-firing of fossil fuel-fired powerplants is not available. In this scenario, the technology cost assumptions are the same as in the Default scenario. In these main scenarios, the socioeconomic conditions were equivalent to the assumptions of Shared Socioeconomic Pathway (SSP) 2[80].

In addition to these main scenarios, sensitivity scenarios were analyzed to evaluate several uncertainties associated with socioeconomic conditions. Although the main scenarios in this study used SSP2 assumptions, SSP1 and SSP3 assumptions were also assessed as sensitivity cases for the 500 H2Opt because the recent IAM-based studies generally included socio-economic uncertainties among SSP 1–3[42,81]. The results of these sensitivity analyses are summarized in the Supplementary Figures.

## IPCC AR6 scenario data

For comparison between our results and the IPCC AR6 scenario data, the World v.1.1 dataset of the AR6 scenario data was used[82]. The categories C1, C2, and C3 were used for comparison, as they were essentially consistent with the range of climate scenarios of this study. C1 limits the temperature increase to 1.5 °C with no or limited overshoot

with 50% likelihood. C2 limits 1.5 °C in 2100 with 50% likelihood after a high overshoot. C3 limits peak warming to 2 °C in this century with 67% likelihood.

## Reporting summary

Further information on research design is available in the Nature Portfolio Reporting Summary linked to this article.

## Data availability

The scenario data generated in this study have been deposited in the Zenodo repository under accession code https://doi.org/10.5281/zenodo.10457307.

## Code availability

The source code used for scenario data analysis and figure production is provided in the GitHub repository (https://github.com/kenoshiro/H2cofire; https://doi.org/10.5281/zenodo.10602980). The source code of the AIM-Technology model is available at the GitHub repository (https://github.com/KUAtmos/AIMTechnology_core; https://doi.org/10.5281/zenodo.8401421).

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

## Acknowledgements

K.O. and S.F. acknowledge support from the Environment Research and Technology Development Fund (JPMEERF20211001) of the Environmental Restoration and Conservation Agency provided by the Ministry of Environment of Japan, JSPS KAKENHI Grant Number JP23K04087, and the Sumitomo Electric Industries Group CSR Foundation.

## Author contributions

K.O. designed the research. K.O. and S.F. contributed to scenario design and interpretation of the results. K.O. developed the model, conducted the analysis, created the figures, and wrote the first draft of the paper. K.O. and S.F. contributed to the final manuscript.

## Competing interests

The authors declare no competing interests.
