## [Peer Review File · Nature Communications]

REVIEWER COMMENTS

Reviewer #1 (Remarks to the Author):

This work reported the energy system model of global power generation with the consideration of hydrogen cofiring. The focus of this work was the effect of hydrogen cofiring in delaying the fossil-fuel generator in different scenarios. The result of the insignificant effect of hydrogen cofiring in delaying the phase-out scenario of fossil fuel generators is novel and interesting. However, there are several points to be clarified and explained further as follows.

1. Integrated assessment models and AIM/Technology employed in this work, including the scope and confidence level, need further explanation. The validation reported from other studies may be added to the manuscript to further support the result and analysis provided in this work.
2. In order to delay the phase-out of fossil fuels power plants, hydrogen cofiring should be utilized as much as possible, i.e., by utilizing the hydrogen cofiring on baseload power generators. Why the hydrogen cofiring in this work is used as a backup source of RE?
3. When assuming the hydrogen cofiring as balancing power or backup of VRE, is there any difference in investment cost required for battery storage?
4. Although in the introduction section, Asian countries are mentioned as the focus in stranded power plant analysis, the results do not much explain about the energy transition of Asian countries. Please explain further, specifically for the energy transition effect in Asian countries.
5. In the discussion section, it is stated that the impact of hydrogen cofiring on prolonging fossil fuel power plants is limited. Also, the absence of hydrogen cofiring is not significant in the energy transition process. These are based on the assumption of the main utilization of hydrogen in a low-capacity factor power generator. Please consider if hydrogen is used in the base load, and the fossil fuels remain used for backup power plants.

Reviewer #2 (Remarks to the Author):

This manuscript reports an energy policy study exploring the global impact of hydrogen co-firing on the power generation sector. The energy system model AIM/Technology was used to perform the quantitative analysis of scenarios. The conclusion is drawn that hydrogen co-firing has limited impact on total power generation and is also limited in extending fossil-based power generation. Generally, this work provides some message for the usage of future hydrogen energy, especially in fossil-based power generation. However, there are some problems on the novelty and analysis which make this manuscript unsuitable for publication at its current stage.

1. About the novelty. The conclusion obtained from this work that the hydrogen co-firing has limited impact on fossil-based power generation has already been mentioned in several publications, while the points differing from these publications were not well mentioned in the manuscript. In fact, future power generation is more and more accepted to be low carbon, and hydrogen energy is considered to compensate the temporal and spatial imbalance of renewable primary energies like solar energy and wind energy. Nevertheless, it is not economical to burn hydrogen in a very large scale, especially in the power generation sector. Therefore, it can readily derive the conclusion that hydrogen co-firing can hardly play an important role in prolonging fossil-based power generation under low emissions scenarios.

2. The energy system model AIM/Technology adopted to perform the quantitative analysis of scenarios was also a well adopted model. No substantial update was made in this work to this model, instead of simple application and update in parameters.

3. To simulate the seasonal storage of hydrogen, 12 representative days per month and 24 hours per day were selected in the AIM/Technology modeling, which seems to be quite arbitrary. What are the criterions of the authors to determine the representative days?

4. A phenomenological analysis of the comparisons between various technologies and related solutions in Figures 1 and 2 is provided, while less in-depth analysis and thoughts are provided behind these comparisons.

5. The figures in this manuscript and in supplementary materials are rather rough.

We appreciate the reviewers' comments and the opportunity to submit a revised manuscript. The manuscript was revised thoroughly according to all comments. This document contains point-by-point responses to each comment in blue. All changes in the revised manuscript are highlighted in yellow.

Reviewer #1 (Remarks to the Author):

This work reported the energy system model of global power generation with the consideration of hydrogen cofiring. The focus of this work was the effect of hydrogen cofiring in delaying the fossil-fuel generator in different scenarios. The result of the insignificant effect of hydrogen cofiring in delaying the phase-out scenario of fossil fuel generators is novel and interesting. However, there are several points to be clarified and explained further as follows.

1. Integrated assessment models and AIM/Technology employed in this work, including the scope and confidence level, need further explanation. The validation reported from other studies may be added to the manuscript to further support the result and analysis provided in this work.

Response:

We appreciate this comment and concur regarding the importance of model validation. Several methods have been utilized for the validation of the process-based models such as AIM/Technology, including diagnostic study, historical simulation, sensitivity analysis, model comparison, and the provision of model documentation, rather than statistical validation presenting the confidence levels of the results (Wilson et al., Ref 61). The following text has been added to describe how the AIM/Technology model provides this information for model evaluation.

Line 408

“There are various methods for model evaluation of process-based IAMs⁶¹. Model intercomparison has been conducted with a number of IAMs^{1, 62}, and some of the models have included model documentation^{63, 64}. In some studies, historical simulations^{65, 66} and diagnostic studies^{67, 68} have been conducted. For the AIM/Technology model, detailed model documentation, including mathematical equations and parameter assumptions, is available⁶⁹. Model intercomparisons have been conducted for national studies^{70, 71, 72}. A recent study using the AIM/Technology-Global model presented a comparison with the IPCC-AR6 scenario³².”

We have also added model and scenario evaluations to the extent possible, based on the results of our study.

First, to evaluate the scope of the model results, an additional sensitivity analysis was performed

by changing both key and uncertain input parameters. The H2Opt+ scenario was added to explore how much hydrogen co-firing is implemented as much as possible, in addition to the existing technology scenarios (LimCCS, H2Opt and NoCOF). A description and a summary of the results is provided in our response to the next comment.

In addition, because socio-economic assumptions can affect general model results, existing IAM studies often consider diverse socio-economic scenarios to examine how the scope of the results may be changed. Although all scenarios in the initial submission of this article were based on SSP2 socio-economic assumptions, we also examined SSP1 and SSP3 assumptions, as has been performed in recent IAM studies (Gambhir et al., Ref 77, Guo et al., Ref 78). Descriptions of these sensitivity scenarios and their related assumptions have been added to the Methods as follows.

Page 4, line 127

“Sensitivity scenarios were also analyzed for various socio-economic conditions.”

Page 14, line 456

“In addition to these main scenarios, sensitivity scenarios were analyzed to evaluate several uncertainties associated with socio-economic conditions. Although the main scenarios in this study used SSP2 assumptions, SSP1 and SSP3 assumptions were also assessed as sensitivity cases for the 500-H2Opt because the recent IAM-based studies generally included socio-economic uncertainties among SSP 1–3^{77,78}. The results of these sensitivity analyses are summarized in the Supplementary Information.”

As an example, the share of hydrogen co-firing in the sensitivity scenarios was shown as follows (Figure S6c). We confirmed that the share of hydrogen co-firing remained below <2% of the total power generation in these sensitivity scenarios. In other cases, different technological and socio-economic assumptions resulted in various energy demand levels and associated energy supply requirements. Nevertheless, we consider that the key findings of our study are robust. We have added text regarding the sensitivity scenarios as follows.

Page 5, line 156

“Nevertheless, the share of power generation associated with hydrogen co-firing is negligible compared to total power generation, accounting for <1% in 2050, including that for sensitivity scenarios with various technological and socio-economic conditions (Figure 1c).”

Supplementary Fig. 6. Share of hydrogen co-firing in the power sector, including the sensitivity scenarios. a Hydrogen co-fired power generation as a share of total thermal power generation. b Fossil fuel-fired power generation, including hydrogen co-firing, as a share of total power generation. c Hydrogen co-fired power generation as a share of total power generation.

Second, model intercomparison is also used as a tool for evaluating IAMs (Wilson et al., Ref 61). Therefore, we compared the scenarios of this study with those of IPCC-AR6 where possible, as in our previous work (Oshiro et al., Ref 32). We mainly compared the fossil fuel phase-out trend and economic indicators (e.g., Figure 1b and Supplementary Figure 2b for example), because the IPCC-AR6 scenarios do not provide hydrogen-related variables. The Results section has been edited as follows.

Page 5, line 146

“Power generation from fossil fuel-fired generators, which includes hydrogen co-firing, decreases rapidly, accounting for around <20 EJ per year by 2050 (Figure 1b). These values are similar to the ranges obtained from the corresponding mitigation scenarios of the IPCC-AR6.”

Page 6, line 190

“It should be noted that the power capacity values in this study are larger than those in the IPCC-AR6 scenarios, as shown in Figure 2b, because the AIM/Technology model has detailed time resolution, which results in a large requirement for back-up capacity for VRE intermittency.”

Figure 1 b) Power generation from fossil fuel-fired generators (including hydrogen co-firing and CCS). Right bar plots illustrate the power generation from fossil fuels in 2050 as obtained from the IPCC AR6 Scenario Database ⁷⁹ for each climate category.

Supplementary Fig. 2. Emissions and carbon prices. a CO₂ emissions from the energy supply and demand sectors. Emission total includes the negative emissions by DACCS. b Carbon prices. Right bar plots illustrate the values in 2050 as obtained from the IPCC AR6 ⁷⁹. Right bar plots illustrate the capacity of total fossil fuel-fired generators in 2050 in the IPCC-AR6. “n” denotes the number of available scenarios in each category. The range of 0-800 US\$ per t-CO₂ is illustrated.

Descriptions of comparisons with AR6 scenarios are provided in the revised Methods section, as follows.

Page 14, line 463

“IPCC AR6 scenario data

For comparison between our results and the IPCC AR6 scenario data, the World v.1.1 dataset of the AR6 scenario data was used⁷⁹. The categories C1, C2, and C3 were used for comparison, as they were essentially consistent with the range of climate scenarios of this study. C1 limits the temperature increase to 1.5°C with no or limited overshoot with 50% likelihood. C2 limits 1.5°C in 2100 with 50% likelihood after a high overshoot. C3 limits peak warming to 2°C in this century with 67% likelihood.”

2. In order to delay the phase-out of fossil fuels power plants, hydrogen cofiring should be utilized as much as possible, i.e., by utilizing the hydrogen cofiring on baseload power generators. Why the hydrogen cofiring in this work is used as a backup source of RE?

Response:

Thank you for this helpful comment. First, in terms of the model structure, the operation conditions for a hydrogen co-fired plant (as for all flexible generators) are not predefined as backup generators for peak load. Rather, the operating conditions are endogenously determined in the model given the supply and demand conditions and the costs. As shown in Figure 2c, because the capacity factor of a natural gas generator with CCS ranges from 10% to 60%, all fossil fuel-fired generators can potentially be operated as both base- and peak-load generators. The following sentences have been added in the Methods section.

Page 11, line 355

“The power sector uses a dispatch module with a 1-h temporal resolution for representative days. The operating conditions of the power generator are determined under a cost-minimizing framework. Therefore, the capacity factor of flexible generators, including hydrogen co-fired generators, is determined endogenously in this model.”

Second, we agree that it is necessary to examine the conditions in which hydrogen co-firing is utilized as much as possible. In the revised manuscript, we added the “H2Opt+” case, which is the most optimistic scenario for hydrogen co-firing. The scenario description has been edited as follows.

Page 13, line 446

“The H2Opt+ scenario was designed as a what-if scenario assuming the most optimistic conditions for hydrogen co-firing. In addition to the conditions imposed in the H2Opt case, this scenario includes higher battery cost assumptions and no new construction of seasonal storage (CAES and PHES). Also, the same limitations on CCS and bioenergy that occur in the LimCCS case were imposed in the H2Opt+ case.”

As shown in Fig. 1c (see below), the share of hydrogen co-fired generators remains limited in the H2Opt+ scenario, because the cost of electricity generation is still higher than that of other generators. Therefore, we consider that the key findings of this paper remain robust for this optimistic scenario. We have following text to the Discussion.

Page 9, line 283

“According to the scenario assessment with various assumptions about mitigation stringency and the technology portfolio, the impact of hydrogen co-firing on prolonging fossil fuel-based power generation is limited. Even in the H2Opt+ scenario in this study, which was designed as a what-if scenario that assumes the most optimistic condition for hydrogen co-firing, the share of hydrogen co-firing accounts for <1% of global total power generation in 2050.”

Figure 1 c, d) Hydrogen co-firing as a share of total power generation in 2030 and 2050 in the world and each region, respectively. e) Hydrogen co-firing as a share of fossil fuel-based power generation relative to carbon prices, excluding NoCOF cases.

3. When assuming the hydrogen cofiring as balancing power or backup of VRE, is there any difference in investment cost required for battery storage?

Response:

Thank you for this helpful comment. Capacity and investment for battery storage have been added to Supplementary Figure 15 as follows. Although investment for battery storage differs among the scenarios to some extent, the difference is smaller than in the case of seasonal storage options, shown in Fig. 3a. Specifically, as the difference between the Default and NoCOF scenarios is very small for battery storage, the effect of hydrogen co-firing itself for battery storage seems moderate. The following sentence has been added to Result section.

Page 7, line 224

“Hydrogen co-firing also has an impact on battery storage capacity with respect to short-term variability (Supplementary Fig. 15); however, it would not reduce the importance of battery as a short-term storage option, because the difference in battery capacity between the default and NoCOF scenarios is relatively small.”

Supplementary Fig. 15. Battery storage. a Battery storage capacity in 2050. b Annual investment for battery storage in 2050. The installation of battery storage does not occur in the H2Opt+ scenario because of the high cost assumptions of this scenario.

4. Although in the introduction section, Asian countries are mentioned as the focus in stranded power plant analysis, the results do not much explain about the energy transition of Asian countries. Please explain further, specifically for the energy transition effect in Asian countries.

Response:

Thank you for pointing this out. As in the initial submission the regional results were summarized only for stranded capacity, we have added results on energy transition by region in this revision. First, in terms of the main results of this paper, the share of hydrogen co-firing and stranded capacity information have been added in the Figures 1d and 4c as follows. We confirmed that regions with greater stranded capacity are characterized by a greater share of hydrogen co-firing. The following text have also been added in Results and Discussion sections.

Page 5, line 163

“Relative to the global average, the share of hydrogen co-firing in total power generation rises in OECD and Asian countries, followed by the Middle East and Africa; however, the contributions of hydrogen co-firing to total power generation remain limited, reaching 2–3% in each region

(Figure 1d).”

Page 7, line 239

“Stranded capacity is observed mainly in Asia, especially for coal-fired generators, followed by OECD countries, the Middle East and Africa (Figure 4c), due to the near-term greater capacity of coal- and gas-fired generators (Supplementary Fig. 18).”

Page 9, line 288

“The hydrogen co-firing results differ among regions; the share of hydrogen co-firing is larger in OECD countries and Asia because of the greater risks of stranded capacity. Nevertheless, the impact of hydrogen co-firing is limited even in these regions, as the share of hydrogen co-firing reaches 2-3% at most.”

Page 7, line 239

“Stranded capacity is observed mostly in Asian regions especially for coal-fired generators, followed by the OECD countries and the Middle East and Africa (Figure 4c, Figure S 16), because of greater capacity of coal- and gas-fired generators in the near-term (Figure S 17).”

Figure 1 c, d) Hydrogen co-firing as a share of total power generation in 2030 and 2050 in the world and each region, respectively. e) Hydrogen co-firing as a share of fossil fuel-based power generation relative to carbon prices, excluding NoCOF cases.

Figure 4. Impacts of hydrogen co-firing on stranded capacity of power generators. c Unused coal and gas power capacities by region for the 500 Gt-CO₂ scenarios. Results for other scenarios are shown in Supplementary Fig. 17.

In addition, regional power generation and primary energy results have been added to the Supplementary Information. The phase-out of fossil fuel is confirmed in all regions, as shown in Supplementary Fig. 1b.

Detailed regional results can be obtained from the spreadsheet referred to in the data availability statement.

Supplementary Fig. 1. Power generation. b Power generation by region in the 500 Gt-CO₂ budget scenarios.

5. In the discussion section, it is stated that the impact of hydrogen cofiring on prolonging fossil fuel power plants is limited. Also, the absence of hydrogen cofiring is not significant in the energy transition process. These are based on the assumption of the main utilization of hydrogen in a low-capacity factor power generator. Please consider if hydrogen is used in the base load, and the fossil fuels remain used for backup power plants.

Response:

Thank you for this comment. As noted in our response to the second comment of Reviewer 1, we included the condition in which hydrogen co-firing is used for a base load generator as much as possible in the H2Opt+ scenario. Even in this scenario, hydrogen co-firing is limited in power generation in a cost-effective framework. Therefore, we consider this finding to be robust. We have added the following sentence to the Discussion.

Page 9, line 285

“Even in the H2Opt+ scenario in this study, which was designed as a what-if scenario that assumes the most optimistic condition for hydrogen co-firing, the share of hydrogen co-firing accounts for <1% of global total power generation in 2050.”

Reviewer #2 (Remarks to the Author):

This manuscript reports an energy policy study exploring the global impact of hydrogen co-firing on the power generation sector. The energy system model AIM/Technology was used to perform the quantitative analysis of scenarios. The conclusion is drawn that hydrogen co-firing has limited impact on total power generation and is also limited in extending fossil-based power generation. Generally, this work provides some message for the usage of future hydrogen energy, especially in fossil-based power generation. However, there are some problems on the novelty and analysis which make this manuscript unsuitable for publication at its current stage.

1. About the novelty. The conclusion obtained from this work that the hydrogen co-firing has limited impact on fossil-based power generation has already been mentioned in several publications, while the points differing from these publications were not well mentioned in the manuscript. In fact, future power generation is more and more accepted to be low carbon, and hydrogen energy is considered to compensate the temporal and spatial imbalance of renewable primary energies like solar energy and wind energy. Nevertheless, it is not economical to burn hydrogen in a very large scale, especially in the power generation sector. Therefore, it can readily derive the conclusion that hydrogen co-firing can hardly play an important role in prolonging fossil-based power generation under low emissions scenarios.

Response:

We appreciate this comment. We agree that several papers have mentioned the role of hydrogen; however, we feel that the following issues have not been clarified sufficiently.

- Previous studies focused only on the role of substituting natural gas-fired generators, ignoring the effect on coal plants despite today's continued increase in coal power generation.
- In previous studies, regional coverage was limited to Europe or the US, or to individual plant-scale comparisons. Therefore, the global impact of hydrogen co-firing, including in Asian countries, where coal generation is notably increasing, remains unclear.
- The stranded asset risk is considerable for regions, such as Asia, but previous studies have not focused on its impacts.
- Condition of hydrogen co-firing in the power sector is unclear. Although some studies have considered the recent cost decline of renewables, an assessment of an optimistic assumption on hydrogen is required.

Therefore, we have reorganized the Introduction to highlight these current knowledge gaps, and added the findings of the most recent studies. In terms of the coal coverage and regional coverage, the following text has been added to the Introduction.

Page 2, line 57

“Nevertheless, as only a few studies have assessed hydrogen use in the power sector, there are still several knowledge gaps associated with the potential role of hydrogen. First, previous studies have mainly focused on hydrogen co-firing only with natural gas-fired generators and with a limited regional coverage. Öberg et al.¹⁸ used a power system model for European countries and found that hydrogen co-firing is limited benefit in natural gas powerplant, even in a stringent mitigation scenario. Bui et al.²² evaluated the CO₂ emission intensity of hydrogen-based generators and indicated that it can contribute to decarbonizing natural gas generators such as CCS. In national model intercomparison studies for Europe and the US, some models have included hydrogen-based energy carriers, concluding that hydrogen is mainly used in high-density transport fuels and high-temperature industrial processes, whereas its use in the power sector is limited^{23,24}. Although these studies have provided insights into the role of hydrogen, the potential of hydrogen co-firing at the global level, including at coal and at gas powerplants, is not yet well understood.”

Regarding the potential impacts on the stranded asset risk, the following sentences have been added.

Page 3, line 70

“Second, although retrofitting fossil-fueled generators with hydrogen co-firing can theoretically reduce stranded asset risks, little is known about such effects. Because global coal power generation is currently still rising, especially in Asian countries²⁵, stranded asset risk can be a significant issue in these regions^{10, 26, 27}. Although previous studies have assessed the effect of retrofitting coal powerplant with carbon capture and biomass co-firing^{28, 29}, the potential impact of hydrogen co-firing on avoiding stranded asset risks has not yet to be explored.”

Regarding the technological conditions, we have assessed an additional scenario that assumes the most optimistic technological assumptions regarding hydrogen co-firing.

Page 4, line 122

“The H₂Opt+ scenario is a what-if scenario with the most optimistic conditions for hydrogen co-firing. In addition to the conditions in the H₂Opt and LimCCS cases, the H₂Opt+ scenario includes higher cost assumptions for battery storage and no new construction of seasonal storage.”

In line with the additional novelty statement on regional diversity, we added several regional results to highlight the novelty of this study, shown, for example, in Figure 1d and 4c as follows.

Figure 1 c-d) Hydrogen co-firing as a share of total power generation in 2030 and 2050 in the world and each region, respectively. e) Hydrogen co-firing as a share of fossil fuel-based power generation relative to carbon prices excluding NoCOF cases.

Figure 4. Impacts of hydrogen co-firing on stranded capacity of power generators. c Unused coal and gas power capacities by region for the 500 Gt-CO₂ scenarios. Results for other scenarios are shown in Supplementary Fig. 17.

2. The energy system model AIM/Technology adopted to perform the quantitative analysis of scenarios was also a well adopted model. No substantial update was made in this work to this model, instead of simple application and update in parameters.

Response:

Thank you for this helpful comment. Although hydrogen production and consumption in the industry, transport, and buildings sectors have already been modelled in previous studies, we consider that substantial updates have been made in this study to analyze hydrogen use in the power sector. Because the model advancements were briefly mentioned in the initial submission, we have added more detailed descriptions in the main text as follows.

First, the revised manuscript explicitly describes the achievement of previous model versions and the new contribution of the present model as follows.

Page 3, line 90

“Although hydrogen production and use in the final energy sectors have been modelled in previous studies ^{14, 32}, there are three major advancements associated with hydrogen applications in the power sector to address the research question of this study.”

In terms of hydrogen co-firing technologies, which are newly considered in this study, detailed descriptions have been added.

Page 3, line 94

“It is assumed that both hydrogen and ammonia can be co-fired for these generators (hereafter, hydrogen co-firing is defined as including both hydrogen and ammonia). Additional investments required for hydrogen co-fired generators, which vary with the hydrogen mix rate, are assumed based on the literature ¹⁸.”

We have also explicitly noted that technology retrofit is endogenized in the model. We consider this to be a unique characteristic among the several IAMs. The revised text is as follows.

Page 4, line 98

“Second, the model was updated to represent the retrofitting of existing fossil fuel-fired generators with hydrogen co-firing and carbon capture technologies. Consequently, a technology retrofit is endogenously determined in the model, based on the additional investment required for the retrofit,

the changes in energy and emissions performances, and the remaining lifetime of the upgraded plant.”

Regarding the temporal resolution, we substantially updated the AIM/Technology model, such that it is now comparable with other power system models. The following text has been added.

Page 4, line 105

“Third, to consider the seasonal variation in electricity demand and supply from VREs, the intra-annual temporal resolution was improved in this model. As previous studies assessing seasonal storage modeled monthly power supply and demand profiles^{35, 36}, the AIM/Technology in this study also contains 12 representative days corresponding to one per month and 24 h per day.”

3. To simulate the seasonal storage of hydrogen, 12 representative days per month and 24 hours per day were selected in the AIM/Technology modeling, which seems to be quite arbitrary. What are the criterions of the authors to determine the representative days?

Response:

We appreciate this comment. The criteria of the representative day choice were based on insights from previous studies. Several detailed energy system models, such as TIMES model, use seasonal or monthly representative days. We also note that time resolutions based on 2-3 representative seasons are insufficient. A clear advantage of selecting seasonal or monthly representative days to assess the seasonal storage effect is that these days represent seasonal or monthly specific electricity supply and demand profiles. Therefore, we selected 12 representative days, one for each month. The following text has been added in Methods section.

Page 12, line 380

“To account for the seasonal variation in electricity demand and supply from VREs, the intra-annual temporal resolution was improved in this model version to include 12 representative days, corresponding to one for each month, and 24 h per day, whereas the previous model used two typical days to represent year, corresponding to summer and winter¹⁴. Due to tradeoffs between the model temporal resolution and computational resource requirements⁴¹, there have been few presentations of the intra-annual temporal resolution in the IAMs. Recently, some IAMs have modeled the seasonal specific electricity supply and demand profiles explicitly^{39, 42}, or soft-linked the IAMs and detailed power system models^{43, 44, 45}. By contrast, detailed power and energy system models, such as The Integrated MARKAL-EFOM System (TIMES), present a more

detailed temporal resolution for four seasons or for each month ^{46, 47, 48}. Although there are several methods to reduce the time resolution for power system analysis ^{49, 50}, the representative day approach for each season or month offers a clear advantage in assessing seasonal storage effects, as it represents seasonal specific supply and demand conditions. Because the 2- to 3-season average approach is insufficient to capture the duration curves of solar and wind ⁵¹, and several studies assessing electricity storage modeled monthly power supply and demand profiles ^{35, 36, 52}, the updated model contains 12 representative days, one for each month, and 24 h per day.”

Although we consider the representative choice in this study to be reasonable, we provide some caveats related to model choice. The Discussion section has been edited as follows.

Page 10, line 328

“Second, as this study is based on single model assessment, it would be useful to employ other models with different structures, such as different temporal resolutions. Although the model choice can affect the quantitative result of power generation to some extent, such as the requirement for seasonal storage, the findings of a limited role of hydrogen co-firing on decarbonization is likely to be robust, because the cost disadvantage of hydrogen co-firing would not be greatly affected by the choice of model.”

4. A phenomenological analysis of the comparisons between various technologies and related solutions in Figures 1 and 2 is provided, while less in-depth analysis and thoughts are provided behind these comparisons.

Response:

Thank you for pointing this out. The results on the limited potential of hydrogen co-firing indicated in Fig. 1, and the lower capacity factors of hydrogen co-fired generators, indicated in Fig.2, are associated with the stranded capacity and LCOE results, which are presented in the later parts of the Results section. Therefore, we have reorganized the first paragraph of the Discussion to highlight the descriptions of the phenomena shown in Figure 1 and 2. The revised text is as follows.

Page 9, line 283

“According to the scenario assessment with various assumptions about mitigation stringency and the technology portfolio, the impact of hydrogen co-firing on prolonging fossil fuel-based power generation is limited. Even in the H2Opt+ scenario in this study, which was designed as a what-

if scenario that assumes the most optimistic condition for hydrogen co-firing, the share of hydrogen co-firing accounts for <1% of global total power generation in 2050. The hydrogen co-firing results differ among regions; the share of hydrogen co-firing is larger in OECD countries and Asia because of the greater risks of stranded capacity. Nevertheless, the impact of hydrogen co-firing is limited even in these regions, as the share of hydrogen co-firing reaches 2-3% at most. Although expansion of hydrogen co-fired generator capacity is observed for both coal and gas, the contributions of these changes to total power generation are small, because the hydrogen co-fired generators mainly used as a back-up option for VRE intermittency. It is due to greater cost penalty associated with hydrogen production, relative to the direct use of generated electricity from renewables or energy penalty of CCS implementation. Although hydrogen cost reduction may enhance hydrogen co-firing, the phase-out of fossil fuel-fired generators is a robust trend in the deep decarbonization scenario. These results suggest that the transition risks of fossil fuel-based power infrastructure must be considered when developing climate and energy strategies, even when hydrogen co-firing becomes technically feasible.”

5. The figures in this manuscript and in supplementary materials are rather rough.

Response:

We appreciate this comment. In the revised manuscript, the figures are presented in vector format (.eps) and the resolutions for the supplementary figures has been improved.

REVIEWER COMMENTS

Reviewer #1 (Remarks to the Author):

The authors have effectively addressed all concerns from the initial review. Key points include:

1. Detailed explanation and validation of the integrated assessment models and AIM/Technology, enhancing the study's credibility.
2. Inclusion of the H2Opt+ scenario, offering a deeper understanding of hydrogen co-firing's potential and limitations.
3. Analysis of hydrogen co-firing's impact on battery storage investments, showing its moderate effect.
4. The addition of regional analysis, especially regarding Asian countries.
5. Considering hydrogen in baseload generation in the H2Opt+ scenario reinforces the limited impact of hydrogen co-firing on extending fossil fuel usage.

The manuscript is substantially improved, addressing each review point comprehensively and leaving no further comments.

Reviewer #2 (Remarks to the Author):

The revised manuscript addresses some of my previous comments. However, critical concerns remain and demand the authors' attention.

1. The prediction indicating a wind power contribution of about 50% to power generation by 2050, surpassing solar PV, seems ambitious given the current trajectory of energy technologies. Achieving this would necessitate an exhaustive exploration of wind energy, contrasting with solar PV, which benefits from a more recognized and abundant resource prospect. Can the authors provide supporting references for this prediction?
2. If wind power and solar PV are anticipated to contribute nearly 80% to power generation, managing their intermittent and regional nature will require stability technologies like hydrogen co-firing, as explored in this work. However, the manuscript doesn't address the potential role of fossil fuels co-fired with hydrogen in the 2050 scenario. Will they function as an independent sector in power generation or merely complement renewable energy?

3. The authors analyze the scenario of hydrogen co-firing and conclude its limited role in the 2050 scenario. Would this role become more significant if we consider a scenario involving pure hydrogen combustion?

4. Figure 1a, the authors predict an extremely rapid increase in the share of power generation from wind power and solar PV starting in 2020, alongside a contrasting trend for fossil energy. This contradicts the current status from 2020 to 2023 and the foreseeable future before 2030, especially considering disruptions caused by the pandemic, economic slowdowns, and global political situations affecting renewable energy investments. The authors should reevaluate their model to accurately simulate the current status and make reasonable predictions for the foreseeable future.

We appreciate the reviewers' comments and the opportunity to submit a revised manuscript. The manuscript was revised thoroughly according to all comments. This document contains point-by-point responses to each comment in blue. All changes in the revised manuscript are highlighted in yellow.

Reviewer #1 (Remarks to the Author):

The authors have effectively addressed all concerns from the initial review. Key points include:

1. Detailed explanation and validation of the integrated assessment models and AIM/Technology, enhancing the study's credibility.
2. Inclusion of the H2Opt+ scenario, offering a deeper understanding of hydrogen co-firing's potential and limitations.
3. Analysis of hydrogen co-firing's impact on battery storage investments, showing its moderate effect.
4. The addition of regional analysis, especially regarding Asian countries.
5. Considering hydrogen in baseload generation in the H2Opt+ scenario reinforces the limited impact of hydrogen co-firing on extending fossil fuel usage.

The manuscript is substantially improved, addressing each review point comprehensively and leaving no further comments.

Response: Thank you again for your constructive comments.

Reviewer #2 (Remarks to the Author):

The revised manuscript addresses some of my previous comments. However, critical concerns remain and demand the authors' attention.

1. The prediction indicating a wind power contribution of about 50% to power generation by 2050, surpassing solar PV, seems ambitious given the current trajectory of energy technologies. Achieving this would necessitate an exhaustive exploration of wind energy, contrasting with solar PV, which benefits from a more recognized and abundant resource prospect. Can the authors provide supporting references for this prediction?

Response:

Thank you for this comment. We concur with the challenges associated with large-scale wind

penetration raised by the reviewer. Nevertheless, we consider it valid based on following two points, which are available resource potential and a higher capacity factor of wind power, and a comparison of wind power generation volumes with existing literature.

In terms of the first point, we added the model’s potential assumptions and methods in Supplementary Fig. 22 (provided below). The estimation is based on the works of Pfenninger *et al.* and Staffell *et al.* as detailed in Supplementary Text 2. While it is acknowledged that solar PV potential surpasses that of onshore wind, our study demonstrates that there is sufficient potential of onshore wind (about 600 EJ per year) to fulfill the wind power generation required in this study’s scenarios. Moreover, the average capacity factor of wind, ranging between about 20-40%, significantly surpasses that of solar PV, predominantly below 20%. Lastly, our wind potential assumption is consistent with the ranges presented in existing studies, such as the 350-1800 EJ per year range stated by Deng *et al.*

Supplementary Fig. 22. Potential for wind and solar power as a function of average capacity factor.

Regarding the second point, we concur with the reviewer’s comment that recent trends in cost decline and capacity additions of solar PV could contribute to heightened expectations for its future prospects. Despite slight modifications to our results due to reassessment of near-term conditions (refer to our response to the fourth comment), the results indicate that wind power generation continues to surpass that of solar PV in 2050. However, we consider that solar and wind shares are uncertain, as this largely hinges on the choice of the model and its associated assumptions.

In alignment with the IPCC AR6 scenarios, we observed a wide range in the share of solar and

wind power. Notably, most IMPs exhibited a higher proportion of wind power generation compared to solar PV, as illustrated in Supplementary Fig 23 (provided below). While some AR6 scenarios may not fully account for the latest developments in solar PV, investigations utilizing models such as GCAM and MESSAGEix reveal a consistent trend of wind generation surpassing that of solar PV in 2050 (e.g., Iyer et al., Guo et al.).

Recognizing the uncertainty inherent in forecasting the solar-wind energy balance, we have expanded the Discussion section as follows.

Page 10, line 334

“In addition, although this study indicates that extensive wind power generation exceed the power generated by solar PV in 2050, which is consistent with recent scenario assessments using the GCAM and MESSAGEix^{41, 42}, expectations of increased solar PV are growing given recent developments. Although the model choice and structure can affect the quantitative result of power generation to some extent, such as the requirement for seasonal storage and solar PV expansion, the findings of a limited role of hydrogen co-firing on decarbonization is likely to be robust, because the cost disadvantage of hydrogen co-firing would not be greatly affected by the choice of model.”

Supplementary Fig. 23 Comparison of the solar and wind share in power generation. Black plots represent the present study’s scenarios with Default technology assumptions. Grey and red circles represent the IPCC-AR6 scenarios in C1-C3 categories and the Illustrative Mitigation Pathways (IMPs)⁸², respectively.

2. If wind power and solar PV are anticipated to contribute nearly 80% to power generation, managing their intermittent and regional nature will require stability technologies like hydrogen co-firing, as explored in this work. However, the manuscript doesn't address the potential role of fossil fuels co-fired with hydrogen in the 2050 scenario. Will they function as an independent sector in power generation or merely complement renewable energy?

Response:

Thank you for this comment. In this analysis, fossil fuel-fired generators co-fired with hydrogen can serve a dual role; addressing intermittency of renewable energy and functioning as base- or middle-load generators akin to other fossil fuel-fired generators. The operation conditions of such co-fired generators are determined endogenously within the model, similar to the determination process for other fossil-fueled generators. This endogenous determination is to minimize the total energy system costs and is contingent upon the behavior of other power generators. We have included clarifying statements in the Introduction section.

Page 3, line 96

“Since the operating conditions of co-fired generators are determined endogenously in the model similar to other fossil fuel-fired generators, hydrogen co-fired generators can play a dual role in complementing the intermittency of VRE and serving as base or middle-load generators.”

Despite the potential role of hydrogen co-fired generators to address VRE intermittency in this study, our results indicate their limited impact. This is primarily due to the existence of alternative and more influential options for addressing VRE, such as electricity storage and curtailment. We highlighted this point again in the Discussion section as follows.

Page 9, line 310

“Nevertheless, our results underscore that hydrogen co-fired generators have a minor influence on the power generation mix. This is attributed to the presence of other cost-effective options for integrating VRE, including PHES and CAES, battery storage, and curtailment.”

3. The authors analyze the scenario of hydrogen co-firing and conclude its limited role in the 2050 scenario. Would this role become more significant if we consider a scenario involving pure hydrogen combustion?

Response:

Thank you for this comment. In our analysis, the hydrogen mix rate can be dynamically adjusted

endogenously within the model, ranging from 0-100%, allowing for the consideration of pure hydrogen (100% hydrogen mix) combustion. However, even when considering pure hydrogen generators, its impact on prolonging coal and gas generators is limited due to the higher cost of hydrogen. The Method section has been revised to provide further clarity.

Page 11, line 377

“In the context of cost minimization, the hydrogen mix rate was internally determined in the model within 0-100%, with a 100% mix indicating exclusive hydrogen combustion.”

Additionally, the header of Supplementary Table 1 has been corrected from “<100%” to “50-100%”.

4. Figure 1a, the authors predict an extremely rapid increase in the share of power generation from wind power and solar PV starting in 2020, alongside a contrasting trend for fossil energy. This contradicts the current status from 2020 to 2023 and the foreseeable future before 2030, especially considering disruptions caused by the pandemic, economic slowdowns, and global political situations affecting renewable energy investments. The authors should reevaluate their model to accurately simulate the current status and make reasonable predictions for the foreseeable future.

Response:

Thank you for this comment. The rapid changes observed in the near-term energy system were predominantly attributed to the start year of emissions reduction, initially set at 2020. In light of the development observed from 2020 to 2023, we have reevaluated our climate policy assumptions, resulting in a revision where emissions reduction now commences from 2024. The updated information has been included in the Methods section.

Page 13, line 441

“Emissions reduction begins from 2024 in all mitigation scenarios, aligning with the historical emissions trend up to 2023⁷⁷.”

Furthermore, we have extended the calibration period for solar and wind power to 2027, taking into consideration the most recent trends and short-term perspectives, as indicated by IEA. The Methods section now reflects this adjustment.

Page 12, line 389

“To incorporate recent trends in solar PV and wind power capacity expansion, the calibration period for solar and wind capacity has been extended to 2027, based on the IEA’s estimates⁴³.”

Consequently, there have been modifications to power generation mix before 2030, reflected in Fig 1a (provided below), with corresponding updates to all figures and numerical values impacted by this adjustment. Notwithstanding, the influence on other indicators, particularly those associated with the role of hydrogen co-firing, is minimal. For instance, the share of hydrogen co-fired power generation remains less than 1% of global power generation in 2050, as depicted in Fig 1c.

In light of these adjustments, the key findings in this paper concerning hydrogen co-firing remain unchanged. We consider that the results indicate that the short-term energy system changes do not compromise the robustness of the key findings associated with hydrogen co-firing presented in this study.

Figure 1. Contribution of hydrogen co-firing to power generation. a Power generation in the 500, 700, 1000 and 1400 Gt-CO₂ scenarios with Default technology.

References

Deng YY, et al. Quantifying a realistic, worldwide wind and solar electricity supply. *Global Environmental Change* 31, 239-252 (2015).

Guo F, et al. Implications of intercontinental renewable electricity trade for energy systems and emissions. *Nature Energy* 7, 1144-1156 (2022).

Iyer G, et al. Ratcheting of climate pledges needed to limit peak global warming. *Nature Climate Change* 12, 1129-1135 (2022).

Pfenninger S, Staffell I. Long-term patterns of European PV output using 30 years of validated hourly reanalysis and satellite data. *Energy* 114, 1251-1265 (2016).

Staffell I, Pfenninger S. Using bias-corrected reanalysis to simulate current and future wind power output. *Energy* 114, 1224-1239 (2016).

REVIEWERS' COMMENTS

Reviewer #2 (Remarks to the Author):

The authors have addressed my comments and the manuscript can be accepted now.

Reviewer #2 (Remarks to the Author):

The authors have addressed my comments and the manuscript can be accepted now.

Response: Thank you again for your constructive comments.